catalysts

MDPI

*Article*

# Nanocomposite Cathode Catalysts Containing Platinum Deposited on Carbon Nanotubes Modified by O, N, and P Atoms

**Vera Bogdanovskaya *, Inna Vernigor, Marina Radina, Vladimir Andreev and Oleg Korchagin**

A.N. Frumkin Institute of Physical Chemistry and Electrochemistry, Russian Academy of Sciences, 119071 Moscow, Russia; msnoviinna@gmail.com (I.V.); merenkovamarina@mail.ru (M.R.); vandr@phyche.ac.ru (V.A.); oleg-kor83@mail.ru (O.K.)
* Correspondence: bogd@elchem.ac.ru

**Abstract:** Platinum deposited on dispersed materials has so far been the most demanded catalyst for creating cathodes for a wide range of electrochemical power sources. This paper sets out to investigate the effect of carbon nanotube (CNT) modification by O, N, and P atoms on the structural, electrocatalytic, and corrosion properties of the as-synthesized monoplatinum catalysts. The investigated Pt/$CNT_{mod}$ catalysts showed an increased electrochemically active platinum surface area and electrical conductivity, as well as an increased catalytic activity in the oxygen reduction reaction (ORR) in alkaline electrolytes. The improved characteristics of Pt/CNT catalysts are explained by alterations in the composition and number of groups, which are formed on the CNT surface, and their electronic structure. By the sum of the main characteristics, Pt/$CNT_{HNO3+N}$ and Pt/$CNT_{HNO3+NP}$ are the most promising catalysts for use as cathode materials in alkaline media.

**Keywords:** monoplatinum catalyst; carbon nanotube; modification; electrochemically active surface area (EAS); oxygen reduction reaction (ORR); corrosion stability

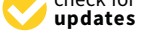



**Citation:** Bogdanovskaya, V.; Vernigor, I.; Radina, M.; Andreev, V.; Korchagin, O. Nanocomposite Cathode Catalysts Containing Platinum Deposited on Carbon Nanotubes Modified by O, N, and P Atoms. *Catalysts* **2021**, *11*, 335. https://doi.org/10.3390/catal11030335

Academic Editor: Carlo Santoro

Received: 12 February 2021
Accepted: 1 March 2021
Published: 5 March 2021



## 1. Introduction

Oxygen electroreduction is one of the most important reactions due to its wide application and demand for practical use in fuel cells (FC) and metal-air power sources [1]. The process of cathodic oxygen reduction limits the characteristics of fuel cells, since the reaction proceeds with a significant overvoltage reaching 300 mV relative to the equilibrium oxygen potential of the $O_2/H_2O$ system. In this regard, it is necessary to create catalytic systems that would meet the requirements of both high activity and selectivity (oxygen reduction directly to water), and stability in harsh operating conditions (oxygen atmosphere, elevated temperatures, and solution pH).

Platinum deposited on a dispersed carbon support (Pt/C) is widely used as a catalytic material for the oxygen reduction reaction (ORR) [2]. Such systems provide high activity; however, despite the excellent corrosion resistance of metallic platinum, Pt/C catalysts degrade during long-term tests in FC due to exposure to corrosive environments [2,3]. Therefore, an important research direction consists in a search for methods capable of improving Pt-containing catalysts not only in terms of their catalytic activity, but also its corrosion stability.

The degradation of Pt/C catalysts [2,3] is associated with a number of reasons, the primary of which is the dissolution of Pt nanoparticles. Although Pt is thermodynamically stable to dissolution at different pH across a wide range of potentials, the Pourbaix diagram shows that Pt dissolves at potentials above 0.85 V (reversible hydrogen electrode, RHE) at pH values below 2, i.e., under the operating conditions of electrodes in low-temperature hydrogen-air FCs [2]. The second reason is related to the corrosion of the carbon support, which causes destruction of the porous structure thus preventing the mass transfer of

reagents towards the active centers and initiating the shedding of Pt nanoparticles. The third reason for reducing the electrochemically active surface area of platinum is the enlargement of its particles due to agglomeration and Ostwald ripening [3].

The development of effective support materials is a priority task on the way of commercialization of FCs, since such materials significantly affect the cost, productivity, and durability of the catalyst, and hence the entire system [4,5]. The following requirements are imposed on the material of catalyst supports [6,7]:

- High specific surface area;
- Predominantly mesoporous structure ensuring the transport of reagents;
- High stability in alkaline and acidic environments under the operating conditions of FCs;
- High proton and electronic conductivity: the electrically conductive material of the support provides a path through which the transfer of electrons between the support and the deposited metal occurs;
- Strong interaction between the nanoparticles of the metal and support material, which increases the stability of the catalyst.

It has been found that promising support materials include carbon nanotubes (CNTs) [8] and new carbon materials with a high surface area, a sufficient number of anchoring sites (active centers), a high electrical conductivity, and resistance to degradation [3]. CNTs exhibit unusual mechanical properties, such as high strength and low density (1100–1300 $kg/m^3$), stability in acidic and alkaline media, as well as a large specific surface area due to the developed porosity [8]. However, in the initial state, the relatively inert surface of these materials is characterized by an insignificant number of active centers necessary for capturing metal nanoparticles [2,6]. As a result, numerous studies are aimed at modifying the CNT surface in order to improve its activity and electronic properties, which is crucial when creating electrocatalysts.

The main approaches to the creation of active centers consist in the formation of oxygen-containing functional groups [9,10] or the introduction of N, P, B, and S heteroatoms into the CNT structure [11–15]. The nature of active groups and their amount may alter the nucleation kinetics and size of metal nanoparticles. This provides a uniform distribution and a smaller size of the nanoparticles of the metal phase [16–18]. The higher the number of active centers, the smaller the size of the Pt particles and the stronger their bond with the support. This facilitates charge transfer thereby increasing catalytic activity [19,20].

The presence of heteroatoms in the CNT structure increases the bond strength of metal with the support as a result of the altered CNT electronic structure, thus increasing the stability of the entire system [17,21]. It was shown [22] that an increase in the nitrogen content leads to a decrease in the size of Pt nanoparticles. Thus, the catalysts containing 1.5, 5.4, and 8.4 at% of nitrogen are characterized by the following sizes of Pt nanoparticles: $5.0 \pm 2.3$, $4.8 \pm 2.0$, and $4.2 \pm 1.7$ nm, respectively. Higher nitrogen contents in the support structure provide a larger number of active centers, smaller sizes of Pt nanoparticles, and larger $S_{EAS}$ values. It was also noted that the stability of the catalyst directly depends on the number of active centers on the support surface. Thus, a Pt catalyst containing 1.5 at% of N preserves only 20.2% of the initial EAS after 4000 cycles. At the same time, the catalysts containing 5.4 at% and 8.4 at% of N preserve 26.6% and 42.5% of EAS, respectively. The authors of [22] explain this increase in stability by an increase in the binding energy between Pt and carbon atoms to 1.35 eV, with an increase in the number of N atoms. However, the decrease in the size of Pt nanoparticles, which actually leads to an increase in the binding energy, was ignored, and no true values of the platinum surface for all the studied systems were presented. The degradation of the support itself, the stability of which increases with an increase in the degree of nitrogen doping, was also disregarded. This fact also justifies the assumption about an increased stability of catalysts on nitrogen-doped CNTs, which was confirmed in a number of works [23–25]. It should be noted that the smallest platinum nanoparticles (d < 2 nm) are the first to dissolve, which is consistent with the

data of [22]. Following corrosion testing, the size of nanoparticles increases, with the most stable catalyst having the smallest nanoparticle diameter of 7.4 nm.

Therefore, the use of CNTs with a modified surface as a support is one of the most promising approaches to increasing the overall activity and stability of catalysts. The abovementioned advantages of modified CNTs allow the Pt loading in the catalyst to be reduced, thus making such catalysts more commercially attractive.

In this work, we aim to establish the effect of the nature and amount of modifying CNT groups on the catalytic and corrosive properties of synthesized on them monoplatinum catalysts in ORR.

## 2. Results and Discussion

Table 1 shows the structural characteristics of the synthesized catalysts. All the catalytic systems have a large mesoporous surface, which is more than 55% of the BET specific surface area for nitrogen-modified CNTs and reaches 100% in the case of a catalyst synthesized on phosphorus-doped CNTs. The mesoporous structure is more favorable than the microporous structure, since it facilitates the diffusion of reagents to the catalytic sites. The largest BET surface area corresponds to the $Pt/CNT_{HNO3+N}$ catalyst. It can also be noted that the least amount of platinum was deposited on CNTs without acid treatment, probably due to the insignificant number of active centers. In acid treated CNTs, the mass fraction of Pt increased and reached ~19 wt.%, reaching 21.2 wt.% after nitrogen doping. Doping with phosphorus, on the contrary, led to a decrease in the Pt mass fraction in the synthesized catalyst. It can be assumed that Pt does not bind to phosphorus atoms on the surface, while phosphorus replaces nitrogen, blocks oxygen and possible defects in the structure of nanotubes.

**Table 1.** Structural characteristics of the synthesized monoplatinum catalysts.

| No. | Catalyst | $S_{BET}$, $m^2$/g | Vpore, $cm^3$/g | Sme, $m^2$/g |
|---|---|---|---|---|
| 1 | 16.3 wt.% $Pt/CNT_{initial}$ | 171 | 0.59 | 130 |
| 2 | 18.75 wt.% $Pt/CNT_{HNO3}$ | 192 | 0.89 | 178 |
| 3 | 21.2 wt.% $Pt/CNT_{HNO3+N}$ | 213 | 0.51 | 119 |
| 4 | 17.8 wt.% $Pt/CNT_{HNO3+P}$ | 161 | 0.99 | 161 |
| 5 | 15.83 wt.% $Pt/CNT_{HNO3+NP}$ | 184 | 0.62 | 126 |

Table 2 and Figure 1 show XPS data on the elemental composition of the surface of modified CNTs used as a support, as well as on the content of Pt nanoparticles on the surface of the most active Pt/CNT catalysts.

**Table 2.** XPS measurements of the studied catalysts.

| No. | Catalyst | Element/at.% in CNT | Element/at.% in Pt/CNT |
|---|---|---|---|
| 1 | $Pt/CNT_{initial}$ | O/0.46 | - |
| 2 | $Pt/CNT_{HNO3}$ | O/15.4<br>N/1.2 | O/11.9<br>N/0.75<br>Pt/1.78 |
| 3 | $Pt/CNT_{HNO3+N}$ | O/12.84<br>N/1.98 | O/8.36<br>N/1.6<br>Pt/1.1 |
| 4 | $Pt/CNT_{HNO3+P}$ | O/10.8<br>N/1.0<br>P/0.2 | - |
| 5 | $Pt/CNT_{HNO3+NP}$ | O/10.8<br>N/1.55<br>P/0.4 | - |

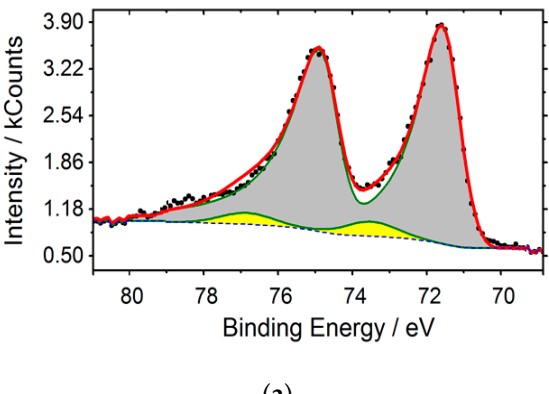
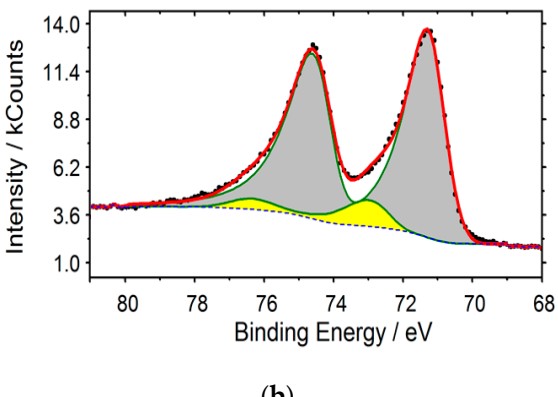

(a)  (b)

**Figure 1.** Pt4f XPS spectra of platinum deposited on carbon nanotubes: (**a**) Pt/CNT$_{HNO3}$; (**b**) Pt/CNT$_{HNO3-N}$.

As can be seen (Table 2), the largest amount of platinum on the surface was found in the Pt/CNT$_{HNO3}$ (sample 2) catalyst, despite the fact that the mass fraction of Pt in this catalyst was lower than in sample 3 (Table 1). It can be assumed that some of the Pt is blocked in the pores, which is indicated by a significant decrease (by almost 50%) in the mesoporous surface area. This value composes 119 m$^2$/g. Moreover, sample 3 has the highest density (Tables 1–3).

**Table 3.** Electrical conductivity of monoplatinum catalysts with a sample weight of 0.015 g.

| No. | Catalyst | Specific Resistance $R$, $\Omega$ cm | Density, g/cm$^3$ | Specific Conductance $\kappa$, S/cm |
|-----|----------|-------------------------------------|-------------------|-------------------------------------|
| 1 | Pt/CNT$_{initial}$ | 5.9 | 1.69 | 0.169 |
| 2 | Pt/CNT$_{HNO3}$ | 4.5 | 1.12 | 0.223 |
| 3 | Pt/CNT$_{HNO3+N}$ | 6.28 | 1.96 | 0.159 |
| 4 | Pt/CNT$_{HNO3+P}$ | 4.7 | 1.00 | 0.212 |
| 5 | Pt/CNT$_{HNO3+NP}$ | 5.5 | 1.61 | 0.182 |

According to XPS data, a decrease in the percentage of O and N atoms on the surface is observed upon the modification of the support material with platinum. This is probably due to their overlapping by Pt nanoparticles when interacting with active centers on the CNT surface. This assumption confirms the fact that the atoms on the CNT surface act as active centers for binding platinum.

### 2.1. Determination of the Electrical Conductivity of Monoplatinum Catalysts

Electrical conductivity is one of the most important characteristics of any electrocatalyst. The electrical conductivity of the catalysts under study was calculated based on the experimentally measured resistance. The data obtained are presented in Table 3.

The specific electrical conductance of the Pt/CNT$_{initial}$ catalyst obtained from the experimental data was 0.169 S/cm. To substantiate the increase in electrical conductivity after functionalization, we present the data on electrical conductivity of CNT$_{initial}$ and CNT$_{HNO3}$, which are 0.066 and 0.097 S/cm, respectively. Upon subsequent modification with platinum, the electrical conductivity of the catalysts more than doubles. It is, for example, 0.223 S/cm for Pt / CNT$_{HNO3}$ versus 0.097 for CNT $_{HNO3}$ (Table 3). When using CNT$_{HNO3}$ as a support for Pt nanoparticles (Pt/CNT$_{HNO3}$), the electrical conductivity increases to 0.223 S/cm. This can be explained by the presence of oxygen-containing groups on the surface of functionalised CNTs, as well as by an increased percentage of Pt on the surface (Table 1, sample 2) and associated increase in density (1.96 g/cm$^3$). The electrical conductivity values for the catalyst synthesized on CNT$_{HNO3+N}$ are significantly lower as compared to Pt/CNT$_{HNO3}$. The first reason for the observed decrease in electrical conductivity may be a lower compression degree of the catalyst during experimental studies. At maximum pressure, the compression degree of the sample depends on its

density: the lower the density, the greater the compression degree and, accordingly, the higher the electrical conductivity. The Pt/CNT$_{HNO3+N}$ catalyst has the highest density and the lowest corresponding compression degree. Under these conditions, the measured resistance is higher, and the electrical conductivity is 1.4 times lower than that of the Pt/CNT$_{HNO3}$ catalyst. At close density values (samples 2 and 4 in Table 3), their electrical conductivities are also close, and some differences are associated with the number and nature of groups on the surface and the mass of deposited platinum (its fraction on the surface). According to the available literature data [26], a high compression degree of a catalytic material based on CNTs leads to an increase in the total area of "van der Waals contacts" between the outer walls of neighboring CNTs, which in turn leads to an increase in electrical conductivity. Second, according to the XPS spectra of Pt4f (Figure 1), the Pt/CNT$_{HNO3 + N}$ catalyst is characterized by a high percentage of Pt in the oxidized form (13.5% Pt-O with a bond energy of 73 eV), while for the Pt/CNT$_{HNO3}$ the proportion of Pt-O is 6% of the total number of Pt atoms on the catalyst surface. This directly affects the value of electrical conductivity. Similarly, the Pt/CNT$_{HNO3+NP}$ catalyst is characterized by a high density and, in addition, a lower percentage of Pt as compared to Pt/CNT$_{HNO3}$ (Table 1), which leads to the low value of its electrical conductivity. The Pt/CNT$_{HNO3+P}$ and Pt/CNT$_{HNO3}$ catalysts have similar characteristics of electrical conductivity: a slight decrease (sample 3) in electrical conductivity is associated with a decrease in the percentage of Pt (Table 1). This indicates that the presence of P atoms in the structure of the carrier material does not affect the electronic conductivity of the catalyst.

## 2.2. Determination of the Electrical Conductivity of Monoplatinum Catalysts

Cyclic voltammetry (CV) measurements (Figure 2) were obtained on Pt catalysts in alkaline and acidic electrolytes. Modified CNTs with a high specific surface area, due to the large number of active centers represented by O-, N-, and P-functional groups, provide the formation of highly dispersed Pt nanoparticles. This leads to an increase in $S_{EASPt}$.

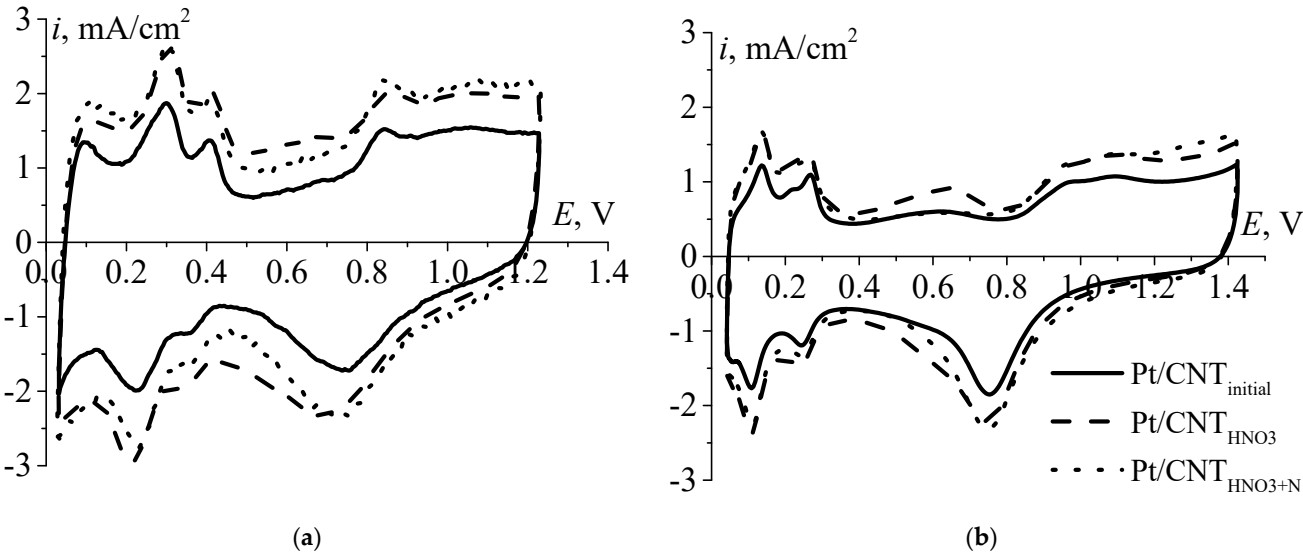

**Figure 2.** Cyclic voltammetry (CV) on Pt catalysts: (**a**) 0.1 M KOH; (**b**) 0.5 M H$_2$SO$_4$. O$_2$; the potential scan rate is 100 mV/s; $m_{kat}$ = 0.15 mg/cm$^2$.

According to Figure 2 and Table 4, the smallest $S_{EAS Pt}$ corresponds to the catalyst synthesized on CNT$_{initial}$. In the initial CNTs, only a small amount of oxygen-containing groups (active centers) is located on the surface of large Pt particles with a small surface area formed during the synthesis. The process of functionalization is associated with the emergence of oxygen-containing groups on the CNT surface, which create new active centers for metal binding and formation of smaller platinum particles. This process is

accompanied by an increase in $S_{EAS\,Pt}$, which is typical of $Pt/CNT_{HNO3}$. Subsequent doping leads to a further increase in the mass of platinum in the catalyst, thus increasing its surface. It can be observed that the platinum load in the catalyst and the value of its $S_{EAS}$ depend on the number of active centers on the support surface. The more active centers are formed, the more platinum is contained in the catalyst, the larger the platinum surface becomes. However, the total percentage of active centers (of all types of modifying atoms: O, N) in the structure of $CNT_{HNO3}$ is the highest among the modified materials amounting to 16.6 at%. About 15% of these centers are oxygen-containing groups on the surface (Table 2), which allows this type of CNT to preserve 18.7% of platinum. For comparison, after doping with nitrogen, 21.2% of the metal is preserved in $Pt/CNT_{HNO3+N}$. Therefore, the presence of predominantly oxygen-containing groups does not provide favorable conditions for Pt deposition. The introduction of nitrogen atoms into the CNT structure increases the number of active centers on the surface and, as a consequence, leads to an increase in $S_{EAS\,Pt}$ (Table 4). The largest number of active centers, exceeding $S_{EAS\,Pt}$ for a commercial catalyst, is observed for $Pt/CNT_{HNO3+NP}$ in acidic and alkaline electrolytes. Thus, for the deposition of highly dispersed Pt nanoparticles on the CNT surface, the presence of nitrogen or nitrogen and phosphorus atoms in their structure is of great significance.

**Table 4.** Electrochemical characteristics of monoplatinum catalysts in acidic and alkaline electrolytes.

| Material/No., as in Table 3 | Pt, wt% | 0.5 M $H_2SO_4$ | | | | 0.1 M KOH | | | |
|---|---|---|---|---|---|---|---|---|---|
| | | $S_{EAS\,Pt}$, $m^2/g$ | $E_{1/2}$, V | $i_{kin}$, mA /cm$^2$ at 0.95 V | $j_{mass}$, mA/mg$_{Pt}$ at 0.90 V | $S_{EAS\,Pt}$, $m^2/g$ | $E_{1/2}$, V | $i_{kin}$, mA /cm$^2$ at 0.95 V | $j_{mass}$, mA/mg$_{Pt}$ at 0.90 V (n) |
| | | | | 1500 rpm | | | | 1500 rpm | |
| Pt/C$^*$/1 | 20 | 48.5 | - | - | - | 51.7 | 0.88 | 0.50 | 55.7 (3.2) |
| Pt/CNT$_{initial}$/2 | 16.3 | 41.0 | 0.83 | 0.3 | 57.75 | 38.3 | 0.85 | 0.36 | 51 (3.1) |
| Pt/CNT$_{HNO3}$/3 | 18.7 | 40.7 | 0.82 | 0.22 | 56.2 | 41.2 | 0.86 | 0.43 | 51.5 (3.3) |
| Pt/CNT$_{HNO3+N}$/4 | 21.2 | 46.7 | 0.83 | 0.27 | 55.4 | 47.1 | 0.88 | 0.61 | 73.0 (3.6) |
| Pt/CNT$_{HNO3+P}$/5 | 17.8 | 54.6 | 0.85 | 0 | 61.7 | 50.3 | 0.87 | 0.53 | 61 (3.1) |
| Pt/CNT$_{HNO3+NP}$/6 | 15.8 | 55.5 | 0.82 | 0.08 | 61 | 54.0 | 0.87 | 0.43 | 65 (3.4) |

ORR activity was determined by the polarization curves shown in Figure 3. The $E_{1/2}$, $i_{kin}$, and $j_{mass}$ values are shown in Table 4.

The catalytic activity of monoplatinum catalysts increases with an increase in $S_{EAS\,Pt}$ (Table 4). Thus, the smaller $S_{EAS\,Pt}$ value corresponds to the lowest ORR catalytic activity, which is observed for $Pt/CNT_{initial}$. The highest activity corresponds to $Pt/CNT_{HNO3+P}$ in an acidic electrolyte and $Pt/CNT_{HNO3+N}$ in an alkaline electrolyte (Figure 3). In an alkaline electrolyte, the $i_{kin}$ values increase proportionally to $E_{1/2}$, while, in an acidic electrolyte, the highest values for $i_{kin}$ are observed for the $Pt/CNT_{initial}$ and $Pt/CNT_{HNO3+N}$ catalysts. It should be noted that the $E_{1/2}$, $i_{kin}$, and $j_{mass}$ values in 0.1 M KOH exceed the parameters obtained at 0.5 M $H_2SO_4$. This indicates a higher activity of catalysts synthesized on modified CNTs in alkaline media.

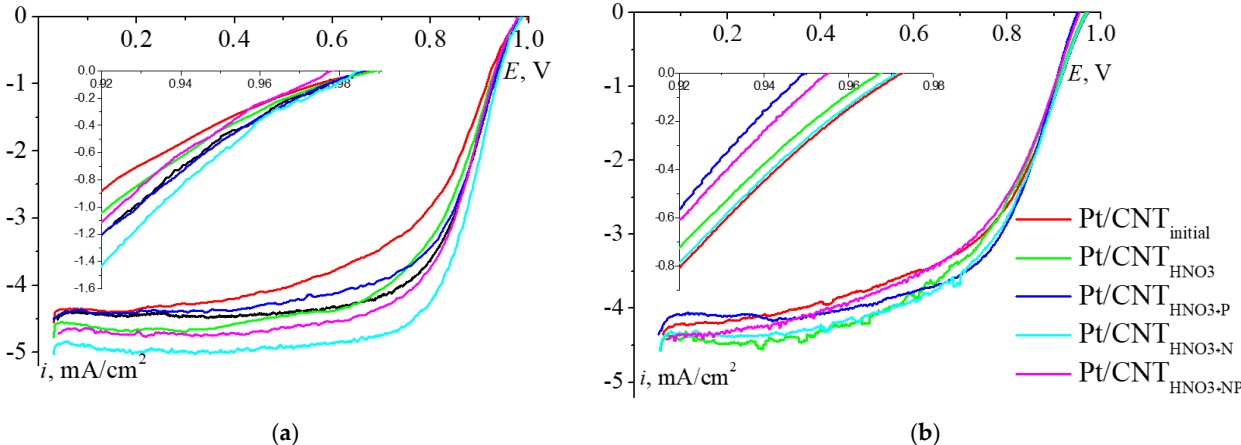

**Figure 3.** Polarization curves on Pt catalysts: (**a**) 0.1 M KOH; (**b**) 0.5 M $H_2SO_4$. $O_2$; the potential scan rate is 5 mV/s; $m_{kat}$ = 0.15 mg/cm$^2$, $w$ = 1500 rpm. The inset shows curve sections in the potential range of 0.98–0.90 V.

In the initial potential range, the $\partial E/\partial \log i$ slopes (Figure 4) have similar values, amounting to about 0.03–0.04 V and 0.08–0.10 V per current decade with increasing polarization. The mechanism of the oxygen reduction reaction on these catalysts does not change depending on the catalyst. In addition, since the values of the slopes are much lower than those observed for the polarization curves obtained on CNTs (more than 0.12 V per current decade [27]) prior to modification, it can be argued that there is an acceleration of the ORR and the rate of electron transfer to an oxygen molecule. This agrees well with the data on the increase in electrical conductivity: the lower slopes of the Tafel dependences correspond to the catalysts synthesized on CNTs, which contain nitrogen and phosphorus atoms in addition to oxygen. Consequently, these catalysts are characterized by the highest electron transfer rate, which confirms their high catalytic activity.

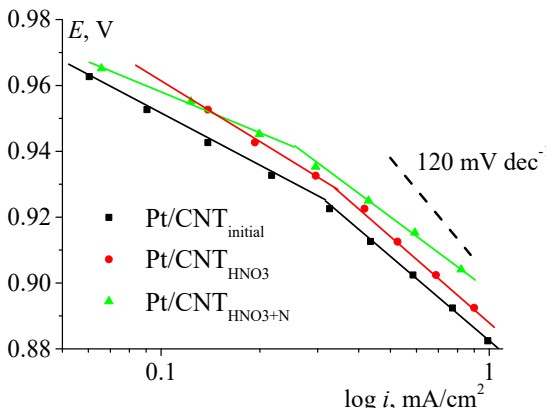

**Figure 4.** Polarization curves in the Tafels coordinates for platinum-modified materials (indicated in the Figure), $O_2$, 0.1 M KOH, 5 mV/s, 1500 rpm.

The value of the limiting current (Figure 5) approaches the theoretically calculated value for the 4-electron process, which comprises 5.6 mA/cm$^2$ at the 1500 rpm electrode rotation speed [10]. This indicates the predominant oxygen reduction directly to water for all the studied systems (Table 4). It can be seen that the oxygen reduction current in the limiting current region depends on the electrode rotation speed in an alkaline electrolyte; therefore, the reaction on the studied catalysts in the limiting current region is limited by oxygen delivery. The current value corresponds (or is close) to the limiting diffusion current of oxygen reduction along the 4-electron path. The highest activity, exceeding that of a 20 Pt/C commercial catalyst, is observed for Pt/CNT$_{HNO3+N}$. This is expressed in higher currents in the whole potential range (Table 4), reaching 0.5 and 0.6 mA/cm$^2$ in the

kinetic region under a similar surface area of platinum, and, in terms of mass activity, at a potential of 0.90 V–55.7 and 73.0 mA/mg$_{Pt}$, respectively.

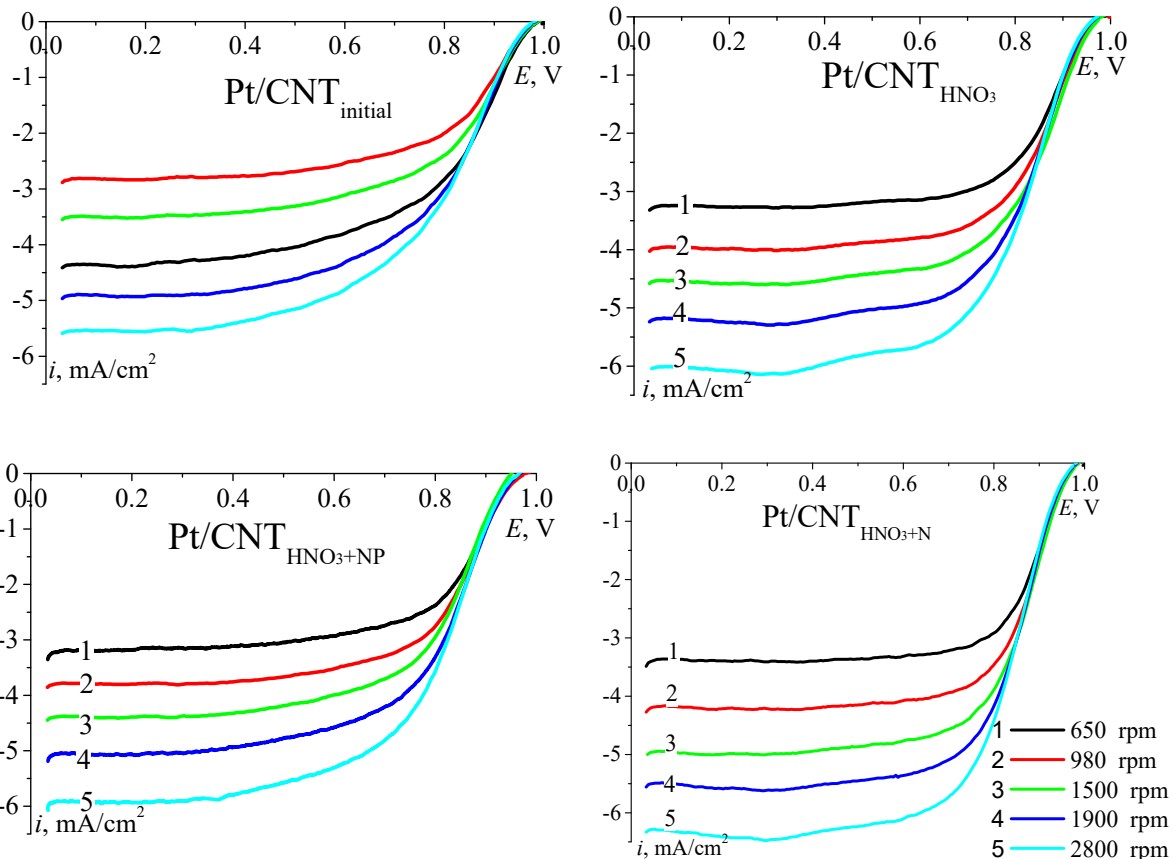

**Figure 5.** Polarization curves recorded on Pt-catalysts at different electrode rotation rates (indicated in the Figure), O$_2$; 0.1 M KOH, 5 mV/s, $m_{um}$ = 0.15 mg/cm$^2$.

### 2.3. Corrosion Stability of Pt-Catalysts

The stability of the catalysts was assessed by the change in $S_{EAS}$ of platinum (Figure 6), i.e., by the decrease in the area of the hydrogen desorption peak, as well as the half-wave potential on the polarization curve of oxygen reduction, after 500 and 1000 cycles.

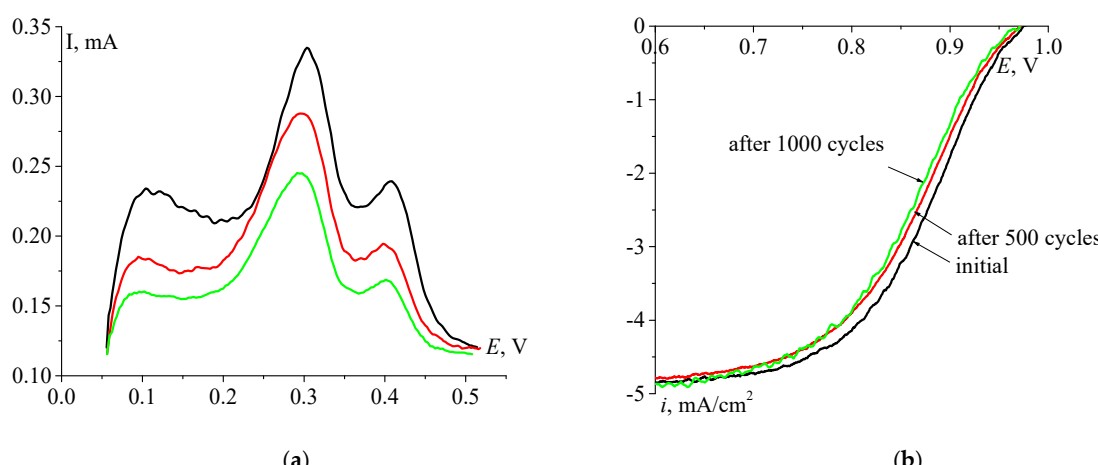

**Figure 6.** (**a**) CV area in the region of hydrogen desorption on Pt/CNT$_{HNO3+N}$, Ar, 0.10 V/s; (**b**) polarization curve recorded on Pt/CNT$_{HNO3+N}$, O$_2$, 5 mV/s, 1500 rpm; initial, after 500 and 1000 cycles, 0.1 M KOH, $m_{kat}$ = 0.15 mg/cm$^2$.

The commercial catalyst is characterized by the lowest stability. Here, the $S_{EAS}$ of Pt is reduced by 60%. The Pt/CNT$_{initial}$ catalyst is characterized by a greater stability due to the low surface area of Pt (large metal particles), which are less prone to dissolution and agglomeration. Among the catalysts synthesized on modified CNTs, Pt/CNT$_{HNO3}$ is the least stable. The decrease in $S_{EAS\,Pt}$ for this catalyst is comparable to Pt/C after 1000 cycles. This result can be attributed to the degradation of the support material, which reduces the overall stability of the system. The Pt/CNT$_{HNO3+N}$ and Pt/CNT$_{HNO3+NP}$ catalysts have similar values of corrosion stability. The greater resistance to degradation of these catalysts is provided by doping (primarily with nitrogen). An increase in the corrosion stability of nitrogen-doped carbon nanotubes was confirmed in a number of works [23–25]. It should also be noted that a large number of oxygen-containing groups and platinum in the oxidized state presented on the surface leads to the destruction of a catalyst. When comparing the XPS spectra of the two Pt/CNT$_{HNO3}$ and Pt/CNT$_{HNO3+N}$ catalysts, along with the same spectra of the CNT$_{HNO3}$ and CNT$_{HNO3+N}$ supports (Table 2), in the first case, a large amount of oxygen-containing groups and a small amount of nitrogen are located on the surface of the support and then on the catalyst based on it. In the second case, there is more nitrogen and less oxygen. In addition, the maxima of the binding energy of platinum electrons in Pt/CNT$_{HNO3}$ are shifted towards higher binding energies (Figure 1), which in fact indicates the presence of platinum in a higher oxidation state considered as the first stage of platinum corrosion. Thus, along with a large number of active centers ensuring the binding of Pt nanoparticles to the CNT surface, it is necessary to use doping atoms of a certain nature, first of all, nitrogen. Additional research is required in order to establish the form of the nitrogen group, which ensures the stability of the catalytic system as a whole. Although the presence of phosphorus or phosphorus and nitrogen simultaneously on the surface provides a developed surface of platinum, the stability of such materials is insufficient (Table 5). It should also be noted that the values of $E_{1/2}$ change insignificantly for all the studied catalytic systems.

**Table 5.** The electrochemical parameters changes of Pt catalysts during accelerated corrosion testing.

| Catalyst | Initial Data | | After 500 Cycles | | After 1000 Cycles | |
|---|---|---|---|---|---|---|
| | $S_{Pt}$, m$^2$/g$_{Pt}$ | $E_{1/2}$, V | $S_{Pt}$, m$^2$/g$_{Pt}$ | $E_{1/2}$, V | $S_{Pt}$, m$^2$/g$_{Pt}$ | $E_{1/2}$, V |
| Pt/C | 60 | 0.88 | 34 | 0.87 | 25 | 0.86 |
| Pt/CNT$_{initial}$ | 38.3 | 0.85 | 33.1 | 0.84 | 24.6 | 0.84 |
| Pt/CNT$_{HNO3}$ | 41.2 | 0.86 | 27.8 | 0.85 | 15.4 | 0.85 |
| Pt/CNT$_{HNO3+N}$ | 47.1 | 0.89 | 33.7 | 0.87 | 25.1 | 0.86 |
| Pt/CNT$_{HNO3+P}$ | 50.3 | 0.88 | 34.3 | 0.88 | 18.9 | 0.87 |
| Pt/CNT$_{HNO3+NP}$ | 54 | 0.87 | 37 | 0.86 | 26 | 0.84 |

## 3. Materials and Methods

Multiwalled CNTs were supplied from Nanotechcenter LLC (Tambov) (>99.0% wt., $S_{BET}$ > 270 m$^2$/g). Melamine (C$_3$H$_6$N$_6$, >99.0%) and triphenylphosphine (TPP) (>99.0%) were purchased from Alfa Aesar (Ward Hill, MA, USA) and Merck (Darmstadt, Germany), respectively. Solutions of NaOH (>97.0%) were purchased from Alfa Aesar. Concentrated HNO$_3$ (70% wt.) for functionalization, KOH (>99.0%), and concentrated H$_2$SO$_4$ (94% wt.) for electrolyte preparation were purchased were purchased from LLC Chimmed (Moscow, Russia). For comparison, commercial (HiSPEC) catalyst 20 wt. % Pt/C was used.

### 3.1. CNT Modification Methods

*Functionalization.* Initial CNTs 500 mg were placed in 100 mL of concentrated HNO$_3$. The mixture was kept at 120 °C for 1 h. After cooling to room temperature, the mixture was washed with deionized water to neutral pH and dried in a vacuum oven at 90 °C. After functionalization, CNTs are designated as CNT$_{HNO3}$.

*Nitrogen doping.* Functionalized CNT $_{HNO3}$ was mixed with melamine in a 1:0.7 ratio and milled in a ball mill (Fritsch Pulverisette 7) for 1 h at 800 rpm. The resulting mixture was heated at 600 °C inside a quartz furnace tube for 1 h in an Ar atmosphere.

*Phosphorus doping.* Functionalized $CNT_{HNO3}$ were mixed with TPP at a 1:5 ratio, dissolved in ethanol, and sonicated for 30 min to form a suspension. The resulting suspension was dried and then heated at 700 °C inside a quartz tube furnace for 1 h in an Ar atmosphere.

*Dual-doping.* To introduce two heteroatoms into the structure of CNTs, two doping techniques were combined. First, CNTs were doped with nitrogen, and then obtained $CNT_{HNO3+N}$ were used as the initial material for doping with phosphorus.

A more detailed description of the functionalization and doping techniques is given in the previous work [23].

### 3.2. The Polyol Synthesis of Monoplatinum Catalysts

This method consists in the reduction of Pt with a polyhydric alcohol at a temperature close to the boiling point. A weighed portion of modified CNTs was mixed with ethylene glycol and subjected to ultrasonic dispersion for 1 h. Subsequently, the CNT suspension in ethylene glycol was placed in a flask. The bath was heated to 130–150 °C followed by a drop-wise addition of a solution containing $H_2PtCl_6$* $6H_2O$ with a concentration of 15 mg/mL to the CNT suspension in the amount required to create a catalyst with 20 wt.% of Pt. The treatment was continued for 1.5 h while bubbling with argon. The resulting mixture was settled down and washed with water; the solid precipitate was separated in a centrifuge and dried.

The Pt content was determined by a spectral method after extracting platinum from the powder with a mixture of nitric and hydrochloric acids and adding $SnCl_2$ to the solution to form a complex with Pt. The intensity of the maximum of the Pt complex (wavelength 401 nm) was used to calculate the platinum content in the sample. For a detailed methodology, see [28].

The following samples were synthesized: $Pt/CNT_{initial}$, $Pt/CNT_{HNO3}$, $Pt/CNT_{HNO3+P}$, $Pt/CNT_{HNO3+N}$, and $Pt/CNT_{HNO3+NP}$.

### 3.3. Electrochemical Methods

Experiments were performed in 0.1 M KOH and in 0.5 M $H_2SO_4$ on a three-electrode system with a glassy carbon electrode (GCE) (0.126 cm$^2$) sealed in Teflon as the working electrode, a Pt foil as the counter electrode, and a Hg/HgO electrode in 0.1 M KOH and an Ag/AgCl electrode in 0.5 M $H_2SO_4$ were used as a reference electrode. Potential values are given versus reversible hydrogen electrode (RHE). All potential values in the article are given relative to a reversible hydrogen electrode (RHE).

To prepare the catalyst ink, 2.2 µg catalysts was dispersed in 500 µL of isopropyl alcohol, 5 µL (~150 µg$_{kat}$/cm$^2$) of this suspension spread on the GCE surface and allowed to dry in air at room temperature.

To determine the electrochemically active surface of studied catalysts in the absence of a depolarizer (oxygen) in the electrolyte solution, cyclic voltammograms (CV) were recorded in the 0–1.2 V (RHE) potential range, at a 100 mV/s scan rate on a stationary electrode. Electrolytes were saturated Ar prior to the start of experiment.

The electrochemically active surface ($S_{EAS}$) of Pt was calculated according to Equation (1) by integrating the charge in the region of hydrogen desorption on CV, taking into account that 0.210 mC/cm$^2$ is necessary for the desorption of a hydrogen monolayer per 1 cm$^2$ of the surface of pure Pt.

$$S_{ESA} = \frac{\int I dE}{0.210 \times v \times m_{Pt}} \tag{1}$$

where *I*—current (A), *E*—the electrode potential (V), *v*—the scan rate (V/s), $m_{Pt}$—the mass of platinum in the active layer on the electrode (mg).

To determine the activity of the studied catalysts in ORR, polarization curves were recorded in $O_2$-saturated electrolyte. Measurements on a rotating disk electrode (RDE) were performed at a 5 mV/s scan rate at a rotation speed varying from 650 to 3000 rpm. Catalytic activity was determined on the basis of half-wave potential ($E_{1/2}$, V), values of the limiting diffusion current density ($i_{lim}$, mA/cm$^2$), mass activity at the potential 0.9 V ($j$, mA/mg$_{Pt}$, and current density in the kinetic region, near the steady-state potential ($i_{kin}$, mA/cm$^2$).

To determine the corrosion stability the method of accelerated corrosion testing was used. During tests, the potential was cycled in the range 0.6–1.3 V in 0.1 M KOH at a potential scan rate of 100 mV/s for 1000 cycles. After 100, 500, and 1000 cycles, changes in $S_{EAS\,Pt}$ and activity in ORR were observed.

### 3.4. Methodology for Measuring Resistance and Calculating Electrical Conductivity

In order to determine the electrical conductivity of dispersed materials, the resistance was measured using a measuring cell with the upper and lower contacts in the form of disks with an area of 0.785 cm$^2$, between which a sample was placed. The initial resistance between the disks, including contact and wire resistance, was 0.015 $\Omega$. A weighed portion (15 mg) of the sample under test was evenly distributed on the surface of the lower disk followed by its pressing by the upper disk. The resistance of the material was measured at a constant (maximum) compression degree. The results of measuring the electrical resistance of materials and the value of electrical conductivity in the form of $1/\Omega$ are presented in Table 3. An assumption was made that, at the maximum compression degree (the distance between the disks in the absence of a powder sample is 0.01 cm), the layer thickness was 0.01 cm when applying the same mass to the disk surface. It is common knowledge that conductivity increases with an increase in the compression degree [26]. In our case, the lower the sample density was, the greater compression was applied. Consequently, the original CNT samples were compressed to a greater extent than Pt-modified samples.

The resistance was measured by the electrochemical impedance method using a Solartron 1287 electrochemical interface. The resistance of the sample was equal to the Ohmic resistance of the zero phases estimated from the impedance spectrum, minus the intrinsic resistance of the cell and wires. The Ohmic resistance of the zero phases is a value cut off on the axis of real resistances by the impedance spectrum. The measurements were carried out in the range 300 kHz–1 Hz, the potential amplitude was 5 mV. The spectra were processed using the ZView software.

In the course of measuring the impedance spectrum, we obtained the dependence of the complex resistance on the frequency of the alternating current. The value of the experimental high-frequency resistance ($R_{ex}$) is equal to the value of the complex resistance, provided that its capacitive component ($Z^{//}$) is equal to zero.

Based on the experimentally measured $R_{ex}$ resistance, the $\rho_{spec}$ specific resistance was calculated according to Equation (2):

$$\rho_{spec} = \frac{Rex \times S}{h}, \tag{2}$$

where $S$ = 0. 785 cm$^2$, $h$ = 0.01 cm is the distance between the disks.

In turn, the $\kappa$ specific electrical conductivity [S/cm] was determined by Equation (3):

$$\kappa = \frac{1}{\rho_{spec}} \tag{3}$$

### 3.5. Structural Studies

*Brunauer–Emmett–Teller (BET) method.* BET surface area ($S_{BET}$) and porosity values of the studied materials were determined using the physical sorption method with a Micrometrics ASAP 2020 setup. The adsorption isotherms were measured at 77 K volumetrically examined by nitrogen gas adsorption.

*X-ray photoelectron spectra (XPS).* XPS were acquired on an Auger spectrometer (Vacuum Generators, UK) with the CLAM2 attachment for measuring XPS spectra. The vacuum in the analyzer chamber was better than 10–8 Torr. An Al anode served as the source of monochromatic radiation (200 W). The peak position was standardized based on the position of a carbon C1s peak with energy of 285.0 eV. For quantitative ratios, we used the coefficients of sensitivity shown in the VG1000 program for spectra processing. The surface layer composition was determined to a depth of 10 nm.

### 4. Conclusions

The modification of CNTs, initially subjected to functionalization, by N and P atoms used as a platinum support produces catalysts, whose activity is comparable to that of a commercial Pt/C catalyst.

Platinum catalysts synthesized on modified CNTs demonstrate a higher activity in alkaline media compared to acidic media. In alkaline electrolytes, $Pt/CNT_{HNO3+N}$ showed the highest activity and stability values among the studied materials, with $E_{1/2}$ = 0.885 V, $i_{kin}$ = 0.61 mA/cm$^2$, and $j_{0.9V}$ = 73 mA/mg$_{Pt}$. In acidic electrolytes, the activity parameters of this catalyst were $E_{1/2}$ = 0.85 V and $i_{kin}$ = 0.27 mA/cm$^2$.

The use of CNTs doped with N and P atoms enhances the stability of platinum catalysts. In terms of general characteristics, the $Pt/CNT_{HNO3+N}$ and $Pt/CNT_{HNO3+NP}$ catalysts are promising for practical use as cathodes in alkaline media.

**Author Contributions:** Conceptualization, V.B.; data curation, V.A.; investigation, I.V. and M.R.; methodology, O.K. All authors have read and agreed to the published version of the manuscript.

**Funding:** This work was carried out with the financial support of the RFBR project BRICS_T No. 19-53-80033.

**Data Availability Statement:** Additional data are available in [23] or upon request from the corresponding author.

**Acknowledgments:** This work was carried out with the financial support of the RFBR project BRICS_T No. 19-53-80033.

**Conflicts of Interest:** The authors declare that they have no conflict of interest.

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
