# Peer review of "Nanocomposite Cathode Catalysts Containing Platinum Deposited on Carbon Nanotubes Modified by O, N, and P Atoms"

_catalysts, doi:10.3390/catal11030335_

Round 1

Reviewer 1 Report

In this investigation, the author modified the CNT structure by O, N, P atoms to study their influence on structural, electrocatalytic, and corrosion properties. Overall, the research idea is very innovative and the results are very promising. However, I have few points for author consideration.

1- Last raw in table 2 hard to understand since I think it has an ordering issue

2- It states that “As can be seen (Table 2), the largest amount of platinum on the surface was found in the Pt/CNTHNO3 (sample 2)” which contracting  table 2 listed Pt/CNT HNO3+N=213 Pt wt% while Pt/CNTHNO3 =192

Author Response

The authors are grateful to the referee for carefully reading the manuscript and for making useful comments.

  1. Table 1 shows data on the structural characteristics of CNTs and Pt / CNTs, which make it possible to establish the relationship between the structure and activity of the synthesized systems. Determine the effect of the preliminary modification of CNTs on the mass of platinum in the composition of the catalyst as a whole. From this group of catalysts, on the basis of electrochemical measurements, samples 2 and 3 were selected that are characterized by higher activity, selectivity, and corrosion resistance and have the highest mass percentage of Pt content (~ 19 and 21wt%) and BET surface area (192 and 213 m2/g). For these samples, additional studies were carried out using the XPS method, which made it possible to explain the effect of the state of platinum on the surface on the characteristics of catalysts in ORR in alkaline electrolytes.

  1. The authors apologize for the mistake made in preparing Table 2. The necessary changes have been made to the table (the second column has been removed, in which the data on the BET surface area for each catalyst were given, and the title indicates that this is the mass content of Pt). All discussions of date of Table 2 are based on the data presented in Tables 1, 2 and 3. Additions and changes have been made to the text (highlighted in yellow).

Reviewer 2 Report

  • Authors should provide a picture of their setup to measure electrical conductivity. The explanation of the procedure to measure electrical conductivity is ambiguous.  As they correctly stated the electrical conductivity will be increased by applying higher pressure. As they stated in the manuscript, they applied different pressure for different samples in accordance with their densities. How can they compare the electrical conductivity of the samples with different compression? authors should be more elaborate on the measurement protocol of the electrical conductivity. Authors should also report the electrical conductivity of the pristine CNT and CNTNH3 to see how this treatment affected the electrical properties of the CNT.
  • Authors should also provide how they measured the density of their samples.
  • Raman measurement of pristine CNT samples with treated samples should be included in the manuscript. 
  • The authors should also provide high-resolution XPS for their samples. For instance, 0.2% of phosphorous owing to the doping is very low and might be just noise or overlapping with other compounds. 
  • No morphological characterization was presented in this work. 

I ask the authors to fully revise their work and apply my comments because it lacks through characterization of their samples

Author Response

In the first part of the review, questions and suggestions relate to the method of measuring the electrical conductivity of synthesized catalysts.

We used a simple device, which is briefly described in the article. It includes two platinum discs, one of which is fixed in the base of the structure and has a down conductor. The second disc (with a down conductor) is fixed on the rod and moves with a screw. Maximum pressure when discs touch. The dispersed sample under study is placed on the lower disk and the upper disk is pressed against it.  

Measurements of electrical conductivity were performed at maximum compression of a sample with the same mass (15 mg). In this case, depending on the specific volume (density), the degree of compression of the sample differed. Thus, we take into account the density of the samples to explain the observed differences. The data obtained are not absolute values ​​of electrical conductivity, but show a tendency for electrical conductivity to change during modification.

To substantiate the increase in electrical conductivity after functionalization, we present the data on electrical conductivity of CNTinitial and CNTHNO3, which are 0.066 and 0.097 S * cm-1, respectively. Upon subsequent modification with platinum, the electrical conductivity of the catalysts more than doubles. These is, for example, 0.223 S * cm-1 for Pt / CNTHNO3, as shown in Table 3. The data are included in the text.

When determining the density of the synthesized catalytic systems, we used the reciprocal of the specific volume, which was obtained by measuring the surface by BET.

Raman spectroscopy is an informative method for identifying groups on the CNT surface, which was not included in the objectives of this work. Although we obtained information on the elemental composition of CNTs and CNTs modified with platinum from XPS data.

The figure below shows the XPS spectra of a catalyst synthesized on nanotubes doped with phosphorus. The figure shows that the spectrum of phosphorus is clearly expressed, and it cannot be attributed to other components. However, its proportion on the surface is insignificant in comparison with other elements.

CNTHNO3 +P

Figure. CNTHNO3 +P XPS spectra: (a) C 1s; (b) O 1s; (c) N 1s; (d) P 2p

Reviewer 3 Report

This manuscript describes the synthesis of nanocomposite cathode catalysts containing platinum deposited CNTs modified by heteroatoms for oxygen reduction reaction. This manuscript contains solid characterizations, and reasonable discussions. Furthermore, the manuscript was well written, and this work can be of great general interest to the broad readers of synthetic metals. Therefore, the reviewer recommends the publication of this manuscript after addressing the following concern.

  • The test in electrocatalytic selectivity of nanocomposite cathode catalysts for oxygen reduction reaction is required to improve the quality of the manuscript. For examples, cyclic voltammetry (CV) or polarization analysis in both N2- and O2-saturated electrolyte can be considered.

Author Response

The authors are grateful to the referee for a positive assessment of our work and for the proposal to provide additional data to strengthen the justification of the presented results.

In fig. polarization curves and CV –curves in oxygen and argon atmosphere are shown.

As can be seen, ORR catalysis is observed on the polarization curves only in an oxygen atmosphere.

CVs in an argon atmosphere give an indication of the state of the electrode surface. In this case, the contribution of the polarization capacity is insignificant.

In our opinion, the introduction of these (additional) data into the article for platinum catalysts is not necessary.

Figure.  Pt/CNTinitial; 0.1 M KOH; mkat= 0.15 mg/cm2; additional information about parameters of electrochemical measurements is indicated in the figure

Round 2

Reviewer 2 Report

Accepted!

Reviewer 3 Report

Now this manuscript can be accepted in Catalysts.